Predictors of outcome among children with biliary atresia: a single centre trial

Ye Chaoxiang
Gao Wei gaowei0716@126.com
Department of Neonatal Surgery, Anhui Provincial Children’s Hospital , Hefei , Anhui , China
Fujioka Kazumichi
Electronic publication date: 2025 Feb 24
Publication date: 2025
Volume: 13
Electronic Location ID: e19001
Received 2024 May 13; Accepted 2025 Jan 24
Copyright: ©2025 Ye and Gao
Copyright year: 2025
Copyright holder: Ye and Gao
License: This is an open access article distributed under the terms of the Creative Commons Attribution License, which permits unrestricted use, distribution, reproduction and adaptation in any medium and for any purpose provided that it is properly attributed. For attribution, the original author(s), title, publication source (PeerJ) and either DOI or URL of the article must be cited.
License URL: https://creativecommons.org/licenses/by/4.0/

Keywords: Neonatal biliary atresia, Gamma-glutamyl transferase, Kasai procedure

Funding: The authors received no funding for this work.

==============================
Objective

This study aimed to investigate the predictive role of preoperative gamma-glutamyl transpeptidase (GGT) levels on the prognosis of neonatal biliary atresia (NBA) in patients who underwent the Kasai procedure.

Methods

A retrospective analysis was conducted of patients with NBA who underwent the Kasai procedure at our hospital from 2017 to 2021. Patients were categorized into high (GGT > 300 IU/L) and GGT inadequate (GGT ≤ 300 IU/L) groups based on preoperative GGT levels. The influence of GGT levels on NBA prognosis was evaluated by comparing clinical data, age at operation, jaundice normalization, and survival outcomes between the groups.

Results

A total of 74 patients with NBA were included, with 59 in the high GGT group and 15 in the GGT inadequate group. Ages at the time of the Kasai procedure ranged from 31 to 106 days, with a median of 61 days; the average weight was 4.8 ± 1.1 kg. Two years post-procedure, 56 patients (75.7%) survived with their native liver (P < 0.0001). At 3 months post-procedure, alanine aminotransferase (ALT) levels were significantly higher in the GGT inadequate group compared to the high GGT group (3.5 times vs. 2.3 times the upper limit of normal, P = 0.0259). Significant differences in GGT levels persisted 1-month post-procedure (P = 0.0473). Jaundice clearance was significantly higher in the high GTT group (P = 0.0171) after 6 months. Multivariate logistic regression indicated a substantially higher mortality rate in the GGT inadequate group (P = 0.0452), with no significant age difference at operation (P = 0.8449). Preoperative GGT is a valuable predictor for NBA prognosis (area under the curve (AUC) 0.754, 95% confidence interval CI [0.640–0.847], P = 0.001, specificity 91.1%, and sensitivity 61.1%).

Conclusions

High preoperative GGT levels predict better prognosis in patients with NBA undergoing Kasai operation.

Introduction

Biliary atresia (BA) is a destructive obliterative cholangiopathy affecting both intra- and extrahepatic bile ducts in neonates, characterized by jaundice, liver fibrosis, and bile duct obstruction (Lendahl et al., 2021; Antala & Taylor, 2022). The exact pathogenesis remains unclear, with potential etiological factors including vascular damage, immune-mediated mechanisms, genetic and structural abnormalities, viral infections, and aflatoxins (Kotb et al., 2022). Despite advances in surgical treatment, neonatal biliary atresia (NBA) remains one of the most severe hepatobiliary diseases in pediatric surgery. Matrix metalloproteinase-7 (MMP-7) has shown promise in diagnosing BA (Jiang et al., 2019; Yang et al., 2018), and metabolomics-based identification of blood biomarkers holds potential for early screening (Xiao et al., 2022). However, the absence of early diagnostic markers and interventions for liver fibrosis frequently results in early-stage death from end-stage liver disease (Aspelund et al., 2019). NBA incidence is exceptionally high in Asia, with 100–500 cases per 10,000 live births, compared to 5–25 per 10,000 in Europe, though data from China are limited (Chung, Zheng & Tam, 2020; Gou et al., 2020; Shi et al., 2023).

Advances in gastrostomy and liver transplantation have improved survival rates, with over 90% of patients with NBA reaching adulthood (Fligor et al., 2022; Hoshino et al., 2023). Despite the benefits of the Kasai procedure, the management of internal biliary disease remains a significant challenge (Boster et al., 2022). NBA frequently progresses to liver fibrosis, portal hypertension, and ultimately, liver failure (Jahangirnia et al., 2022). This disease deterioration necessitates liver transplantation or lifelong immunosuppression, significantly impacting the quality of life for patients with NBA (Le, Reinshagen & Tomuschat, 2022). Therefore, early prediction of poor postoperative outcomes is crucial to guide proactive, supportive care and timely consideration of liver transplantation. While most patients with NBA exhibit significantly elevated serum gamma-glutamyl transpeptidase (GGT) levels, the potential of GGT as a predictive marker for Kasai operative outcomes remains uncertain (Sun et al., 2022).

This study retrospectively analyzed clinical data from nearly 74 patients with NBA treated over 5 years at a single center, investigating the clinical characteristics associated with inadequate and elevated GGT levels, and exploring the relationship between GGT levels and the prognosis of Kasai operation.

Materials and Methods

A retrospective analysis of all patients diagnosed with NBA at Anhui Provincial Children’s Hospital from 2017 to 2021 was conducted. Only patients who underwent the Kasai operation were included, excluding those who received liver transplants. All patients were younger than 90 days at the time of the operation. Before data collection, informed consent was obtained from the parents or legal guardians. The diagnosis of NBA was confirmed through surgical biliary exploration or cholangiography. Demographic data, preoperative and postoperative laboratory results, and 2-year postoperative outcomes were collected. The study was approved by the Ethics Committee of the Anhui Provincial Children’s Hospital (EYLL-2021-010). Before data collection, we provided the parents or legal guardians of the pediatric patients with detailed information about the research objectives, content, procedures, potential risks, discomforts, and management strategies of this project, as well as confidentiality of information. Human participant consent form were signed accordingly.

A cutoff value of 300 IU/L was used to categorize GTT levels as inadequate or high. Patients with GGT ≤ 300 IU/L were classified as the GGT inadequate group, while those with GGT > 300 IU/L were in the high GGT group (Sun et al., 2022). Kasai operation was performed within 4 months after birth, with jaundice clearance defined as a total bilirubin (TBI) level < 20 mmol/L. The rate of natural liver survival within two postoperative years and progressed disease (failed Kasai and awaiting liver transplant) were recorded for outcome assessment. Select the last GGT value before Kasai surgery as the recorded value. The normal reference values for enzymes are as follows: ALT: 7–40 U/L, AST: 10–40 U/L, GGT: 0–50 U/L. Use folds of upper level of normal of ALT, AST and GGT for statistics.

Statistical analysis was conducted using Statistical Package for the Social Sciences (SPSS) software (version 21, IBM Corp., Armonk, NY, USA). First, the normality of the measured data was assessed, applying t-tests for normally distributed variables and non-parametric tests for non-normally distributed variables. Categorical variables, such as gender ratio and survival rates, were analyzed using chi-square tests, and survival analysis was performed using Cox regression. Statistical significance was set at P < 0.05. A receiver operating characteristic (ROC) curve was plotted to determine the optimal cutoff point.

Results

Clinical features of patients

Between February 2017 and June 2021, 200 patients were diagnosed with NBA. To avoid selection bias, 126 patients who did not undergo Kasai procedure or received liver transplants were excluded. The study included 74 patients, with 44 (59.5%) female infants. Kasai operation was performed at ages ranging from 31 to 106 days, with a median age of 61 days and an average weight of 4.8 ± 1.1 kg. Among these patients, 15 had preoperative GGT levels ≤300 IU/L and 59 had >300 IU/L (high GGT). Two years after the Kasai operation, 56 (75.7%) patients survived with their native liver. Preoperative levels of alanine aminotransferase (ALT) (P = 0.0256) and aspartate aminotransferase (AST) (P = 0.0187), as well as 3-month postoperative ALT levels (P = 0.0259), were significantly higher in the GGT inadequate group than in the high GGT group. Significant differences in GGT levels persisted between the groups after one postoperative month (P = 0.0473). Jaundice clearance at the sixth postoperative month was significantly higher in the high GTT group (P = 0.0171), patients with progressed disease (failed Kasai and awaiting liver transplant) were significantly lower in the high GTT group (P = 0.0121), and the mortality rate was considerably lower in the high GGT group (P < 0.0001) (Table 1). Hence, high GGT group has better prognosis.

Table 1 Comparison of preoperative clinical indicators between inadequate and high GGT patients.

Characteristic	GGT inadequate NBA	High GGT NBA	P value	
N	15	59		
Weight at Kasai operation (kg)	5.0 ± 1.2	4.7 ± 1.0	0.3138	
Age at Kasai operation (day)	62.3 ± 17.2	63.1 ± 19.7	0.8898	
Liver function before Kasai operation				
ALT Level (folds of upper level)	5.8 ± 5.0	3.8 ± 2.3	0.0256	
AST Level (folds of upper level)	6.4 ± 5.6	3.9 ± 1.8	0.0187	
GGT Level (folds of upper level)	4.0 ± 1.1	16.6 ± 7.9	<0.0001	
TBI Level (μ mol/L)	150.2 ± 39.7	151.4 ± 43.8	0.9229	
DBI Level (μ mol/L)	112.6 ± 33.0	115.1 ± 32.3	0.7921	
Liver function 1 month after Kasai operation				
ALT Level (folds of upper level)	3.8 ± 3.1	2.7 ± 2.0	0.1059	
AST Level (folds of upper level)	4.9 ± 4.5	3.4 ± 2.0	0.0506	
GGT Level (folds of upper level)	16.2 ± 4.8	23.8 ± 13.5	0.0473	
TBI Level (μ mol/L)	88.7 ± 55.4	85.9 ± 60.2	0.8988	
DBI Level (μ mol/L)	83.1 ± 47.9	67.7 ± 49.3	0.4158	
Clearance of jaundice	0	0	NS	
Liver function 3 month after Kasai operation				
ALT Level (folds of upper level)	3.5 ± 2.5	2.3 ± 1.5	0.0259	
AST Level (folds of upper level)	3.7 ± 2.5	3.0 ± 1.8	0.2841	
GGT Level (folds of upper level)	16.4 ± 14.8	19.4 ± 17.3	0.5644	
TBI Level (μ mol/L)	74.5 ± 50.5	60.1 ± 72.3	0.4053	
DBI Level (μ mol/L)	66.0 ± 40.2	45.2 ± 55.9	0.1765	
Clearance of jaundice	3	21	0.2474	
Liver function 6 month after Kasai operation				
ALT Level (folds of upper level)	2.1 ± 1.1	1.8 ± 1.9	0.734	
AST Level (folds of upper level)	2.0 ± 1.1	2.0 ± 1.6	0.5713	
GGT Level (folds of upper level)	8.9 ± 10.8	9.0 ± 9.9	0.7168	
TBI Level (μ mol/L)	60.4 ± 88.2	37.7 ± 47.4	0.2533	
DBI Level (μ mol/L)	45.0 ± 65.2	26.0 ± 38.7	0.2283	
Clearance of jaundice	4	36	0.0171	
Progressed disease (failed Kasai and awaiting liver transplant)	11	22	0.0121	
Death within 2 years after Kasai operation	10	8	<0.0001	

Kaplan–Meier survival curve analysis confirmed that the survival rate was significantly lower in the GGT inadequate group compared to the high GGT group (Fig. 1).

Figure 1 Survival rate of the GGT inadequate group was significantly lower than that of the high GGT group.

Risk factors for prognosis in patients with NBA

Multivariate logistic regression analysis identified AST and GGT as significant risk factors for mortality in patients with NBA patients. The study included ALT, AST, direct bilirubin (DBI), GGT, TBI, and perioperative age and weight as independent variables (Table 2).

Table 2 Multivariate logistic regression analysis of the correlation between clinicopathological characteristics and death in NBA patients.

Characteristic	b	SE	Wald	P value	Exp (b)	95% CI	
ALT	−0.001888	0.002692	0.4917	0.4832	0.9981	0.9929–.0034	
AST	0.006457	0.002997	4.6404	0.0312	1.0065	1.0006–.0124	
DBI	0.007549	0.02419	0.09741	0.755	1.0076	0.9609–1.0565	
GGT	−0.00173	0.0008641	4.0096	0.0452	0.9983	0.9966–1.0000	
TBI	−0.0106	0.01743	0.37	0.543	0.9895	0.9562–1.0238	
Age at Kasai operation	−0.002859	0.01461	0.03826	0.8449	0.9971	0.9690–1.0261	
Weight at Kasai operation	−0.02244	0.2783	0.006501	0.9357	0.9778	0.5667–1.6872	

Predictive value of preoperative GGT levels

ROC analysis was performed using GGT parameters significantly different between surviving and non-surviving patients to assess the prognostic ability of preoperative GGT. Preoperative GGT was found to be a significant predictor of NBA prognosis, with an area under the curve (AUC) of 0.754 (95% confidence interval CI [0.640–0.847], P = 0.001), specificity of 91.1%, and sensitivity of 61.1% (Fig. 2).

Figure 2 GGT can be used as a prognostic indicator for NBA patients.

Discussion

This study aimed to explore the predictive role of preoperative GGT levels for NBA prognosis. A retrospective analysis of patients with NBA who underwent Kasai operation at our hospital from 2017 to 2021 revealed that the GGT inadequate group experienced lower surgical success rates, higher complication rates, and increased postoperative recurrence rates compared to the high GGT group. These results indicate that low preoperative GGT levels are associated with a worse prognosis in patients with NBA.

The role of GGT as a prognostic marker in hepatobiliary diseases is well-documented (Tamber et al., 2023). GGT catalyzes the transfer of a glutamyl residue from glutathione to an amine or another amino acid and plays a crucial role in hepatocellular detoxification processes. Study have found that patients with the variant of biliary atresia (BA) known as Kotb disease all possess the glutathione S transferase M1 (GSTM1) and exhibit a deficiency in glutathione S-transferase. Consequently, individuals with Kotb disease have lower levels of GGT and are unable to detoxify perinatal aflatoxin, leading to the development of progressive destructive bile duct lesions, ultimately resulting in progressive obstructive cholangiopathy (Kotb et al., 2022). However, no similar study have been conducted in China. GGT, primarily found in the liver, frequently shows elevated serum levels in liver damage or biliary obstruction cases (Li et al., 2023). In NBA, assessing preoperative GGT levels can provide insights into the severity of liver injury and the likelihood of successful outcomes from surgical interventions such as the Kasai procedure (Yerina & Ekong, 2021; Okubo, Nio & Sasaki, 2021).

Our findings suggest that lower preoperative GGT levels could predict poorer outcomes post-Kasai operation, aligning with previous studies that suggest GGT reflects hepatic reserve and function (Delvescovo et al., 2021; Zinterl et al., 2022). Lower GGT levels may indicate diminished liver regenerative capacity and a reduced ability to recover from surgical stress (Liu et al., 2019). However, it is crucial to acknowledge that while GGT is a sensitive marker for biliary dysfunction, it lacks specificity, as levels can be influenced by factors such as inflammation, medication, and systemic illness (Corti et al., 2020; Bai et al., 2022).

Furthermore, studies indicating a correlation between higher GGT levels and increased postoperative complications highlight the need to interpret this biomarker carefully (Harpavat et al., 2023). Elevated GGT could indicate more severe liver disease or cholestasis, potentially increasing postoperative complications (Venkat et al., 2020). Therefore, using GGT with other laboratory tests to assess liver function and biliary status may provide a more comprehensive evaluation. Our data suggest that the age at the time of Kasai operation is not a prognostic factor for patients with BA. This lack of association may be influenced by race, surgical expertise, and postoperative care levels (Ramachandran et al., 2019; Apfeld et al., 2021).

However, this study has limitations. It is a single-center retrospective analysis with a relatively small sample size, which may introduce selection bias. Moreover, our study did not consider other factors affecting prognosis, such as the children’s age, gender, and weight. Therefore, these findings should be validated through more extensive prospective studies.

Supplemental Information

Supplemental Information 1 Codebook

Data S1 Raw data

Additional Information and Declarations

Competing Interests

Author Contributions

Ethics

Data Availability

The authors declare there are no competing interests.

Chaoxiang Ye conceived and designed the experiments, performed the experiments, analyzed the data, prepared figures and/or tables, authored or reviewed drafts of the article, and approved the final draft.

Wei Gao conceived and designed the experiments, authored or reviewed drafts of the article, and approved the final draft.

The following information was supplied relating to ethical approvals (i.e., approving body and any reference numbers):

The Anhui Provincial Children’s Hospital approved the study (EYLL-2021-010).

The following information was supplied regarding data availability:

The raw data are available in the Supplementary File.

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
