# Peer review of "Predictors of outcome among children with biliary atresia: a single centre trial"

_PeerJ, doi:10.7717/peerj.19001_

## Round 0.1 · original submission · Major Revisions

Reviewers have conflicting opinions.

Please respond sincerely to the points raised by the reviewers. (the acceptance or rejection of your manuscript will be reconsidered as a result of re-examination.)

Reviewer 1 ·

Basic reporting

Clear and unambiguous, lisprofessional English shoud be improved

The article has sufficient introduction and background
The structure of the article is acceptable format

Experimental design

Research question well defined, relevant & meaningful.

The submission clearly define the research question

Validity of the findings

Meaningfull finding

Additional comments

Need English impovement

Reviewer 2 ·

Basic reporting

.

Experimental design

.

Validity of the findings

.

Additional comments

The work is important and relevant to physicians managing children with the relentless biliary atresia (BA).

Kindly amend your manuscript to include the following Points:

Major points

Kindly add the normal age reference value for GGT among neonates of same age in China. Kits for GGT levels vary greatly.

Kindly add the rational for using the GGT cut-off of 300IU/L?

Table 1 clearly outlines that the resolution of cholestasis at 3 months post Kasai was the same (p= 0.247). Then 3 months later there was a significant decline among those with low GGT despite the same level of GGT (p=0.716) followed by increased mortality within 2 years.

Hence, the GGT level post-operatively rose to the same GGT among both groups. Why was this decline in outcome observed? Was it related to age at operation? Rate of repeated cholangitis attacks? Please add the number of cholangitis attacks and other factors that might be responsible for this decline, development of cirrhosis, liver cell failure, ascites, etc

Minor points:

Please use past tense in describing your work.

Abstract: Please outline clearly the outcome of those with initial high GGT and those with low GGT on clearance of cholestasis, improvement of cholestasis, progression of disease and mortality, and the p values.

Please add the p value of logistic regression of GGT level and GGT level associated with age at operation on outcome. Please add the sensitivity, specificity of GGT on outcome.

Please present the ALT and AST levels in folds of upper level of normal for age. Please note that the ALT and AST levels kits vary greatly and upper fold of normal varies according to used kits.

Abstract conclusion: please restrict your self t the evidence presented in the results section of the abstract. Kindly note that all the conclusion that you state in the abstract are not supported by the results you presented. Please simply answer the question presented in the aim clearly. Was GGT predictive of outcome or not? Please state clearly if GGT level was the only determinant of outcome in BA in your study or not?

Introduction:

GGT is a detoxification enzyme. You need to present the literature is focusing on determining the toxin that is causing this GGT up regulation and increment.

Please engage the readers in the introduction and cite the literature on aflatoxins, bilatresone, metaloproteinases, metabolomics, etc in BA

Methods:

Please mention the timing of the initial GGT sampling. Was this the peak GGT value before Kasai? Was it the at the 4th week of life was it at the 6th week???

Please present the ALT and AST levels in folds of upper level of normal for age. Please note that the ALT and AST levels kits vary greatly and upper fold of normal varies according to used kits.

Please specify that you resorted to ROC curve in the statistical analysis section.

Lower GGT is already known to predict poor outcome.

The discussion needs to address what your research adds to already known. You need to share your scientific argument. You need to interpret your own results:

why the age at Kasai was not predictive of outcome, etc

Reviewer 3 ·

Basic reporting

Insufficient field background provided.

Experimental design

too simple.

Validity of the findings

ok.

Additional comments

The study aims to investigate the predictive role of preoperative GGT levels on the prognosis of neonatal biliary atresia (BA) in patients undergoing the Kasai procedure. After analyzing data of 74 BA patients from a single center, the authors concluded that low preoperative GGT level is a risk factor for poor prognosis in BA patients undergoing the Kasai procedure. However, this is not a new finding. Several papers had been reported similar findings,such as :1. Sun S, Zheng S, Lu X et al. Clinical and pathological features of patients with biliary atresia who survived for more than 5 years with native liver. Pediatr Surg Int 2018;34: 381–86; 2.Shankar S, Bolia R, Foo HW et al. Normal gamma glutamyltransferase levels at presentation predict poor outcome in biliary Atresia. J Pediatr Gastroenterol Nutr 2020; 70: 350–55; 3. Guotao. Wang , Huadong. Chen, Xiaoyan. Xie, Qinghua. Cao, Bing. Liao, Hong. Jiang, Quanyuan. Shan, Zhihai Zhong, Wenying. Zhou, Luyao. Zhou(*). 2D shear wave elastography combined with age and serum biomarkers prior to kasai surgery predicts native liversurvival of biliary atresia infants. Journal of Internal Medicine.2020 Nov;288(5):570-580.

---

## Round 0.2 · Major Revisions

Please revise carefully following the reviewers comment. The reviewer was not satisfied with the first revision.

Reviewer 2 ·

Basic reporting

Additional comments
Dear Authors.
You have not addressed the previous points raised by the reviewers.
What is the normal value of GGT in your study? is it 22/55/122 IU/L??????
Please use the folds of upper level of GGT through out the statistics of your study.
Figure 2 is not clear at all. What is the sensitivity and specificity measured?
Why did you choose 300 IU/L as as cut-off. Is GGT 300 = 5 folds of upper level of normal? is it normal??
GGT transfers the gamma-glutamyl group of glutathione which is the second superfamily of detoxification. The discussion needs to relate to GGT function. Why was GGT low? was the lack of increase a reason for expediated cirrhosis in response to aflatoxins?

Kindly amend your manuscript to include the following Points:

Major points

Kindly add the normal age reference value for GGT among neonates of same age in China. Kits for GGT levels vary greatly.

Kindly add the rational for using the GGT cut-off of 300IU/L?

Table 1 clearly outlines that the resolution of cholestasis at 3 months post Kasai was the same (p= 0.247). Then 3 months later there was a significant decline among those with low GGT despite the same level of GGT (p=0.716) followed by increased mortality within 2 years.

Hence, the GGT level post-operatively rose to the same GGT among both groups. Why was this decline in outcome observed? Was it related to age at operation? Rate of repeated cholangitis attacks? Please add the number of cholangitis attacks and other factors that might be responsible for this decline, development of cirrhosis, liver cell failure, ascites, etc

Minor points:

Please use past tense in describing your work.

Abstract: Please outline clearly the outcome of those with initial high GGT and those with low GGT on clearance of cholestasis, improvement of cholestasis, progression of disease and mortality, and the p values.

Please add the p value of logistic regression of GGT level and GGT level associated with age at operation on outcome. Please add the sensitivity, specificity of GGT on outcome.

Please present the ALT and AST levels in folds of upper level of normal for age. Please note that the ALT and AST levels kits vary greatly and upper fold of normal varies according to used kits.

Abstract conclusion: please restrict your self t the evidence presented in the results section of the abstract. Kindly note that all the conclusion that you state in the abstract are not supported by the results you presented. Please simply answer the question presented in the aim clearly. Was GGT predictive of outcome or not? Please state clearly if GGT level was the only determinant of outcome in BA in your study or not?

Introduction:

GGT is a detoxification enzyme. You need to present the literature is focusing on determining the toxin that is causing this GGT up regulation and increment.

Please engage the readers in the introduction and cite the literature on aflatoxins, bilatresone, metaloproteinases, metabolomics, etc in BA

Methods:

Please mention the timing of the initial GGT sampling. Was this the peak GGT value before Kasai? Was it the at the 4th week of life was it at the 6th week???

Please present the ALT and AST levels in folds of upper level of normal for age. Please note that the ALT and AST levels kits vary greatly and upper fold of normal varies according to used kits.

Please specify that you resorted to ROC curve in the statistical analysis section.

Lower GGT is already known to predict poor outcome.

The discussion needs to address what your research adds to already known. You need to share your scientific argument. You need to interpret your own results:

why the age at Kasai was not predictive of outcome, etc

Experimental design

Seriously flawed as they did not explain why did they choose the GGT 300 cutoff level

Validity of the findings

--

Additional comments

--

---

## Round 0.3 · Major Revisions

Please revise the interpretation of your data. Please read table 1 thoroughly.

Reviewer 2 ·

Basic reporting

Dear Authors,
You have 32 major confounders that need to be addressed:
1- Please present your statistics as folds of upper level of normal of ALT, AST and GGT.
Please mention clearly the kits you are using for the ALT, AST and GGT. Please mention the normal cutoff value for each kit. Please use folds of upper level of normal in all 3 enzymes.
The kits all over the world differ. The absolute normal values of ALT, AST and GGT vary, hence your work is very very confounded.

2- Please present outcome as resolved cholestasis, improved cholestasis, progressed disease (failed Kasai and awaiting liver transplant) and death. Native liver survival is not enough.

3- Change the conclusion please.
36 of 59 with high GGT cleared the jaundice while only 4/15 with low GGT cleared the jaundice. 8/59 of high GGT died while 10/15 of low GGT died. (Table 1) Hence the numbers say that hight GGT has better prognosis.

Experimental design

1- Please present your statistics as folds of upper level of normal of ALT, AST and GGT.
Please mention clearly the kits you are using for the ALT, AST and GGT. Please mention the normal cutoff value for each kit. Please use folds of upper level of normal in all 3 enzymes.

Validity of the findings

Please revise interpretation of your data.
Please read table 1 thoroughly.

---

## Round 0.4 · Minor Revisions

Please consider rewriting your work in terms of inadequate response of GGT.

Reviewer 2 ·

Basic reporting

.

Experimental design

.

Validity of the findings

.

Additional comments

Dear Authors,
Your work is impressive, yet has very serious confounders and your interpretation and conclusion are skewed.
Please note that the 15 children with GGT <300IUL have 6 fold elevation of GGT and DO NOT have NORMAL GGT.
Normal GGT is a complete misnomer.
Your work cannot be presented as high and low GGT among biliary atresia patients.
I suggest that you rewrite the work to be "Predictors of outcome among children with biliary atresia: A single centre trial".
Please note that those who could not mount enough GGT had the worst outcome.
Please note that the GGT functions as a transport molecule, helping to move other molecules around the body. GGT catalyses the transfer of a glutamyl residue from glutathione to an amine or another amino acid. Glutathione is essential for hepatocelluar detoxification.
Your findings are very much in agreement with the biliary atresia Kotb disease (BAKD) findings among Egyptian neonates. The evidence supports that all those with BAKD have a detoxification defect. They all have glutathione E transferase M Null genotype, and have glutathione S transferase deficiency. Hence all BAKD cannot detoxify perinatal aflatoxins and develop the progressive destructive cholangiopathy that ends in progressive obstructive cholangiopathy.
Please change the name of the group with "Normal" GGT to "inadequate" or less than 10 folds rise to more than 10 folds rise.
Please consider rewriting your work in terms of inadequate response of GGT.
Moreover, GGT is upregulated in response to chemicals that need detoxification. Hence, your work supports that those with inadequate GGT upregulation are prone to more destructive obstructive consequences.
Please consider discussing the possibility that BA in China might be the results of an offending chemical; aflatoxin, biliatresone etc., in those with a detoxification defect.

---

## Round 0.5 · accepted · Accept

The description of the following three points could have been improved but the content is deemed acceptable for publication:

The function and role of GGT
Mention of BAKD (Kotb disease)
Factors related to biliary atresia in China